# Coptidis alkaloids extracted from *Coptis chinensis* Franch attenuate IFN-γ-induced destruction of bone marrow cells

Jinyu Li[1☯], Xiaoying Meng[1☯], Changzhi Wang[1☯], Huijie Zhang[1☯], Hening Chen[1], Peiying Deng[1], Juan Liu[1], Meiyier Huandike[1], Jie Wei[2]*, Limin Chai[1]*

1 Key Laboratory of Chinese Internal Medicine of Ministry of Education and Beijing, Dongzhimen Hospital, Beijing University of Chinese Medicine, Beijing, China, 2 Pharmaceutical Departments, Dongzhimen Hospital, Beijing University of Chinese Medicine, Beijing, China

☯ These authors contributed equally to this work.
* liminchai@hotmail.com (LC); weijie952157@126.com (JW)

**Data Availability Statement:** All relevant data are within the manuscript and its Supporting Information files.

## Abstract

Coptidis alkaloids are the primary active components of *Coptis chinensis* Franch. Clinical and pharmacodynamic studies have confirmed that Coptidis alkaloids have multiple therapeutic effects including anti-inflammatory, antioxidant and antitumor effects, and they are usually used to treat various inflammatory disorders and related diseases. Mouse bone marrow cells (BMCs) were isolated from BALB/c mice. Immune-mediated destruction of BMCs was induced by interferon (IFN) -γ. High-performance liquid chromatography-electrospray ionization/ mass spectrometry was used to analyze the ingredients of the aqueous extract from *Coptis chinensis* Franch. The results confirmed that Coptidis alkaloids were the predominant ingredients in the aqueous extract from *Coptis chinensi*s. The functional mechanism of Coptidis alkaloids in inhibiting immune-mediated destruction of BMCs was studied *in vitro*. After Coptidis alkaloid treatment, the percentages of apoptotic BMCs and the proliferation and differentiation of helper T (Th) cells and regulatory T (Treg) cells were measured by flow cytometry. The expression and distribution of T-bet in BMCs were observed by immunofluorescence. Western blotting analysis was used to assay the expression of key molecules in the Fas apoptosis and Jak/Stats signaling pathways in BMCs. We identified five alkaloids in the aqueous extract of *Coptis chinensis*. The apoptotic ratios of BMCs induced by IFN-γ were decreased significantly after Coptidis alkaloid treatment. The levels of key molecules (Fas, Caspase-3, cleaved Caspase-3, Caspase-8 and Caspase-8) in Fas apoptosis signaling pathways also decreased significantly after treatment with low concentrations of Coptidis alkaloids. Coptidis alkaloids were also found to inhibit the proliferation of Th1 and Th17 cells and induce the differentiation of Th2 and Treg cells; further, the distribution of T-bet in BMCs was decreased significantly. In addition, the levels of Stat-1, phospho-Stat-1 and phospho-Stat-3 were also reduced after Coptidis alkaloid treatment. These results indicate that Coptidis alkaloids extracted by water decoction from *Coptis chinensis* Franch could inhibit the proliferation and differentiation of T lymphocytes, attenuate the apoptosis of BMCs, and suppress the immune-mediated destruction of the BMCs induced by pro-inflammatory cytokines.

**Funding:** This work was supported by Research special for National Chinese Medicine Clinical Research Base of State Administration of Traditional Chinese Medicine of the People's Republic of China (Grant number: JDZX2015191, http://www.satcm.gov.cn/). The funder had no role in study design, data collection and analysis, decision to publish, or preparation of the manuscript. We have uploaded the information of funding on the revised files.

**Competing interests:** The authors have declared that no competing interests exist.

## Introduction

The activation and expansion of self-reactive pathogenic T cells and highly secreted inflammatory cytokines, such as interferon gamma (IFN-γ) and tumor necrosis factor α (TNFα), can impair the signaling of several cytokine receptors, mediate inflammation and target cell destruction. These pathological changes are the key cellular events in the development of immune-mediated bone marrow (BM) failure. Helper T (Th) 1 cells and their response contribute to these negative effects [1–3]. Both the Fas/FasL and perforin/granzyme pathways are involved in cytotoxic T cell-induced targeting of cells for apoptosis, resulting in the immune-mediated BM destruction [4]. Abnormal expression of Fas and FasL interferes with the development of pancytopenia and marrow hypoplasia. The major role of Fas/FasL cytotoxic pathway in immune-mediated BM failure has been verified [5]. IFN-γ, known as a suppressor of hematopoiesis, can activate the Fas-dependent apoptotic pathway and then initiate cell apoptosis of hematopoietic stem cells (HSCs). The protein levels of Fas, caspases and other proapoptotic genes was shown to significantly increased in BM cells induced by IFN-γ [1].

*Coptis chinensis* is the dried rhizome of *Coptis chinensis* Franch [6]. In traditional Chinese medicine (TCM), *Coptis chinensis* has several therapeutic effects, including providing relief for the negative effects of a hot and humid climate and detoxification. It is usually used to treat various inflammatory disorders and related diseases [7]. Pharmacological studies have demonstrated that *Coptis chinensis* has a variety of pharmacological functions, such as neuroprotection [8], anti-atherosclerosis, anti-diabetes anti-inflammation antioxidation and antitumor effects [9]. The main active ingredients of *Coptis chinensis* are Coptidis alkaloids, including magnoflorine, jatrorrhizine epiberberine, coptisine, palmatine and berberine [10]. The effective components of Coptidis alkaloids could regulate the activation of several processes, including the mitogen-activated protein kinase (MAPK) pathway, endoplasmic reticulum stress, phosphatidylinositol 3-hydroxy kinase and oxidative stress [11].

Clinical treatment with TCM usually occurs via oral administration of a water decoction solution. The herbal active ingredients and the mechanisms of herbal treatment are unclear. Here, we identified the dominant components in the aqueous extract of *Coptis chinensis* by high- performance liquid chromatography-electrospray ionization/ mass spectrometer (HPLC-ESI/MS$^n$) analysis. The freeze-dried powder of *Coptis chinensis* extracted by water decoction was used as a drug treatment. Mouse BM cells (BMCs) induced by IFN-γ were used as a cell model. Different concentrations of freeze-dried powder of *Coptis chinensis* powder was used to treate BMCs for two amounts of time. The percentages of apoptotic BMCs and T cell subunits were measured. The expression and distribution of T-bet were observed by immunofluorescence. The expression of key molecules in the Jak/Stat and Fas apoptotic signaling pathways was assayed by Western blotting. In this study, we wanted to assess the inhibitory functions and potential mechanisms of aqueous extract from *Coptis chinensis* in relation to the apoptosis of BMCs induced by pro-inflammatory cytokines.

## Materials and methods

### Preparation of the aqueous extract

Herbal samples of *Coptis chinensis* Franch (Dried rhizome; Lot number: 18121303; Origin: Sichuan, China) were obtained from Beijing Xidan Pharmaceutical Co, Ltd, China. A senior Chinese medicine appraiser (Jie Wei) undertook the formal identification of *Coptis chinensis* Franch. The herbal inspection reports are shown in S1 File. A total of 100 g of raw herbal sample was extracted by boiling water. The decoction components were soaked in water for 30

min, and then decocted into an extract solution (1 mg/mL). The entire filtered solution of the mixture was concentrated under reduced pressure and then dried by a vacuum freeze drier.

## HPLC-ESI/MS$^n$ analysis

The freeze-dried powder of water decoction solution was used for component analysis. The constituents of *Coptis chinensis* were analyzed by HPLC-ESI/MS$^n$. The specific measurement procedures referred to our previous works methods as previously described [12–14]. Peak View Software 2.2 was used to analyze the data, including retention time, accurate mass and MS/MS spectrum comparison.

## Preparation of bone marrow single-cell suspension

Briefly, 8-week-old BALB/c mice (Protocol No. SCKK (Jing) 2019–0006) were sacrificed by pentobarbital anesthesia. The tibias, femurs and humeris of mice were dissected. Both ends of the bones were removed with sharp scissors. Bone marrow was flushed out of the bone cavity using RPMI-1640 (Thermo Fisher Scientific Inc., Waltham, MA, USA) medium supplemented with 10% FBS (Gibco, Grand Island, NY); then, the bone marrow was filtered through a 70-mm filter mesh, washed with PBS, and resuspended. BMCs were then incubated at 37˚C in a 5% $CO_2$ incubator. Mice were purchased from HFK Bioscience Co., Ltd. (Beijing, China). All animals were kept under standard lighting conditions (12 h alternating day and night cycles) and given free access to food and water. Animal care and use were carried out in accordance with institutional guidelines, and all animal experiments were approved by the Institutional Animal Care and Use Committee of the National Institute of State Scientific and Technological Commission.

## Cell Counting Kit-8 (CCK-8) assay

Cell viability was evaluated by CCK-8 assay. Briefly, BMCs were plated in 96-well culture plates ($1 \times 10^5$ cells/well) with RPMI-1640 supplemented with 10% FBS. Subsequently, cells were divided into different groups treated with various concentrations of the herbal aqueous extract (0, 10, 50, 100, 250, 500, 750 and 1000 μg/mL) at 37˚C in a humidified 5% $CO_2$ incubator. After 24 h of incubation, CCK-8 solution (20 μL) was added into each well and then incubated for another 4 h at 37˚C. The optical density of the purple solution was measured at 570 nm by a microplate reader (Model 3550 Microplate Reader, Bio-Rad Laboratories, Inc., Hercules, CA, USA). The results of the CCK-8 assay indicated that the two concentrations (100 μg/mL and 250 μg/mL) of freeze-dried powder were suitable for cell stimulation. In particular, the two concentrations were optimal for reflecting the dependence and correlation between drug intervention time and drug dose (S1 Fig).

## Cell treatment

BMCs were divided into four groups: normal group, incubated in RPMI-1640 containing 10% FBS; model group, incubated in RPMI-1640 containing 10% FBS with IFN-γ (10 ng/mL) (R&D systems, Minneapolis, MN, USA); and *Coptis chinensis* treatment groups, incubated in RPMI-1640 containing 10% FBS with IFN-γ (10 ng/mL) and the aqueous extract of *Coptis chinensis* (100 μg/mL or 250 μg/mL). After 12 h or 24 h of incubation, cells were harvested.

## Fluorescence-Activated Cell Sorter (FACS) analysis

The percentages of apoptotic cells in BMCs were measured using an Annexin V-FITC Apoptosis Detection kit (eBioscience Inc, San Diego, CA, USA). Briefly, cells were washed twice with

precooled PBS, and then were resuspended in 500 μL of binding buffer at a concentration of $1\times10^6$ cells/mL. After adding 5 μL of Annexin V-FITC solution and propidium iodide (PI) (1 μg/mL), the cells were incubated for 15 min at room temperature. The apoptosis rates of BMCs were analyzed by flow cytometry. To quantify the percentages of $CD3^+CD4^+$ cells, BMCs were washed and stained with anti-mouse CD3 PE-Cyanine7 (0.5 μg, $1\times10^5$ cells/test) and CD4 PE-Cyanine5 (0.06 μg, $1\times10^5$ cells/test) antibodies (eBioscience, San Diego, CA, USA). Anti-mouse CD4 PE-Cyanine5, IFN-γ PE (0.25 μg, $1\times10^5$ cells/test), IL-4 PE-Cyanine7 (1 μg, $1\times10^5$ cells/test) and anti-mouse/rat IL-17A FITC antibodies (0.25 μg, $1\times10^5$ cells/test) (BD Bioscience) were used to determine the percentages of cells that were Th1, Th2 and Th17 cells. Anti-mouse CD4 PE-Cyanine5, anti-mouse CD25 PE (0.125μg, $1\times10^5$ cells/test) and anti-mouse/rat-FOXP3-FITC antibodies (1 μg, $1\times10^5$ cells/test) (eBioscience) were used to determine the percentages of Treg cells. Flow cytometry was performed using a Facs Calibur cytometer. CellQuest software (Beckman Coulter, Brea, CA, USA) was used to analyze the data.

## Immunofluorescence

After 12 or 24 h of incubation, BMCs were washed with PBS, and fixed with 4% paraformaldehyde solution for 10 min at 4˚C. After washed with PBS twice, cells were permeabilized by 0.2% Triton X-100 in PBS for 30 min. After washing, cells were blocked with 1% BSA for 30 min. Then, cells were incubated with anti-human/mouse T-bet PE (1:100) (eBioscience) to examine the expression and distribution of T-bet in BMCs. Images were captured with a Leitz/ Leica TCSSP2 microscope (Leica Lasertechnik GmbH, Heidelberg, Germany). At least 50 cells from 3 different areas of each chamber were measured.

## Western blotting analysis

The protein expression levels of caspase-3, cleaved caspase-3, caspase-8, cleaved caspase-8, Fas, Stat-1, and Stat-3 and the phosphorylation of Stat-1, and Stat-3 were analyzed by Western blotting. After 12 h or 24 h of treatment, BMCs were washed with PBS three times and lysed in RIPA buffer with protease inhibitor cocktail. The protein samples were separated by 10% sodium dodecyl sulfate polyacrylamide gel electrophoresis and then were electrotransferred to nitrocellulose membranes. The membranes were incubated with 1:1000 dilutions of anti-mouse caspase-3, caspase-8, cleaved caspase-3, cleaved caspase-8, Fas, Stat-1, Stat-3, phospho-Stat-1, and phospho-Stat-3 rabbit monoclonal antibodies (CST, Boston, MA, USA) at 4˚C overnight. Then, the membranes were incubated with a horseradish peroxidase (HRP)-conjugated anti-rabbit secondary antibody (CST) for 1 h at room temperature. The immunoreactive proteins were detected with Super Signals west Pico Chemiluminescent Substrate (Thermo Scientific, Rockford, IL, USA). Densitometry plots showing protein expression were normalized to β-actin.

## Statistical analysis

SPSS13.0 software (SPSS Inc, Chicago, IL, USA) was used for the statistical analyses. The data in this study are provided as the means ± standard deviation (S.D.). Normality tests and homogeneity of variance tests were conducted. The data satisfied a normal distribution and homogeneity of variance, and single-factor analysis of variance (ANOVA) was used. The Tukey-Kramer test for multiple comparisons was used to compare the groups. A $P$ value of $<0.05$ was considered statistically significant.

## Results

### Identification of the chemical constituents in aqueous extract of *Coptis chinensi*s

Five constituents were identified based on the accurate mass and relative ion abundance of the target peaks by HPLC-ESI/MS$^n$. The identified compounds and corresponding molecular formulas are shown in Fig 1 and Table 1. All five components are alkaloids. The dominant component in *Coptis chinensis*'s water extract was berberine. Other alkaloids (magnoflorine, jatrorrhizine, epiberberine and coptisine) made up smaller proportions of the aqueous extract of *Coptis chinensis*.

### Coptidis alkaloids in the aqueous extract of *Coptis chinensis* attenuated the cell apoptosis of the IFN-γ-induced BMCs

As shown in Fig 2, the apoptosis ratios of BMCs were increased significantly after 12 h and 24 h of IFN-γ induction compared with those in the normal group ($P<0.01$). The percentages in the treatment groups were decreased significantly as compared with those in the model group after 24 h of treatment with Coptidis alkaloids ($P<0.01$). There was no significant difference between the two drug concentrations (100 μg/mL and 250 μg/mL) after 12 h of treatment. As shown in Figs 3 and 4, the protein levels of Fas, caspase-3, cleaved caspase-3, caspase-8 and cleaved caspase-8 in the model group were increased significantly compared with those in the normal group ($P<0.01$). The levels of Fas in the treatment groups decreased significantly compared with those of the model groups ($P<0.01$). The levels of caspase-3 and caspase-8 in the Coptidis alkaloid treatment groups decreased significantly compared with those in the model group after 12 h and 24 h of treatment ($P<0.01$ or $P<0.05$). Furthermore, the abnormal cleaved activation of caspase-3 and caspase-8 was also inhibited by Coptidis alkaloids. The inhibitory functions of Coptidis alkaloid treatment at 250 μg/mL concentration were superior to those of other groups.

### The distributions of T-bet and the percentages of CD3$^+$CD4$^+$ T cells in BMCs after treatment

As shown in Fig 5, the number of T-bet$^+$-stained cells increased significantly in BMCs induced by IFN-γ (12 h, $P<0.05$). Coptidis alkaloid treatment significantly decreased the quantity of T-bet$^+$ cells in BMCs at 100 μg/mL after 12 h of treatment. The inhibitory effect of Coptidis alkaloids at 250 μg/mL after 24 h of treatment was superior to that of other groups.

Flow cytometry analysis showed that the percentages of CD3$^+$CD4$^+$ T cells increased significantly in the treatment groups compared with the model group after 12 h of treatment ($P<0.01$ or $P<0.05$). Interestingly, the proliferation of CD3$^+$CD4$^+$ T cells in the treatment groups after 24 h of treatment was significantly lower than that of the model group ($P<0.01$). These results showed that the regulatory function of Coptidis alkaloids on the proliferation of CD3$^+$CD4$^+$ T cells was time and dose dependent (Fig 6).

### Effects of Coptidis alkaloids on the proliferation and differentiation of Th cells and regulatory T (Treg) cells in BMCs induced by IFN-γ

The percentages of CD4$^+$IFN-γ$^+$ Th1 and CD4$^+$IL-17$^+$ Th17 cells were increased significantly comprised with those in the normal groups (12 h $P<0.01$ or 24 h $P<0.05$) (Figs 7 and 8), and the percentages of CD4$^+$IL-4$^+$ Th2 and CD4$^+$CD25$^+$FOXP3$^+$ Treg cells were decreased significantly compared with those of the normal groups after 12 h and 24 h of Coptidis alkaloid treatment ($P<0.01$ or $P<0.05$) (Figs 9 and 10). Coptidis alkaloid treatment downregulated the

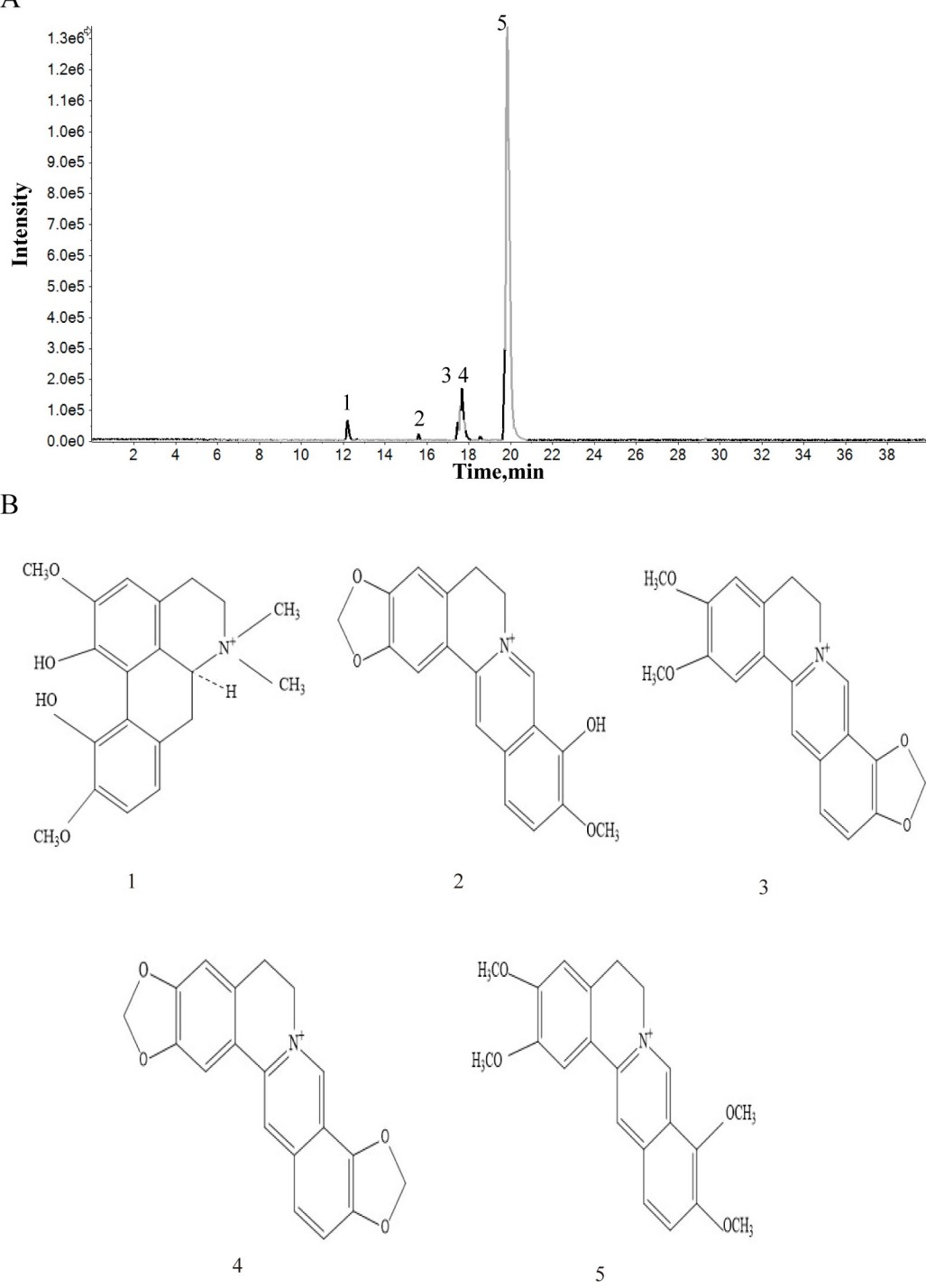

**Fig 1. Characteristics of pure compounds from the aqueous extract of Coptis chinensis.** (A) HPLC-ESI/MS[n] base peak of the total ion chromatograms for the aqueous extract of lyophilized powder of *Coptis chinensis*. The abscissa represents the retention time, and the ordinate represents the chromatographic peak intensity. (B) The molecular formulas of identified components.

differentiation of Th1 and Th17 cells and upregulated that of Th2, and Treg cells ($P<0.05$ or $P<0.01$, compared with those of the model groups). Interestingly, the percentage of Th17 cells

**Table 1. Chemical components identified from *Coptidis chinensis* by HPLC-ESI/MS$^{n}$.**

| Peak | tR(min) | Formula | Identification | Peak Area | Area % |
|---|---|---|---|---|---|
| 1 | 12.21 | $C_{20}H_{24}NO_4$ | Magnoflorine | 3801000 | 3.8% |
| 2 | 17.47 | $C_{20}H_{20}NO_4$ | Jatrorrhizine | 2919000 | 2.2% |
| 3 | 17.6 | $C_{20}H_{18}NO_4$ | Epiberberine | 6043000 | 4.6% |
| 4 | 17.69 | $C_{19}H_{14}NO_4$ | Coptisine | 10440000 | 7.9% |
| 5 | 19.85 | $C_{20}H_{18}NO_4$ | Berberine | 93800000 | 80.5% |

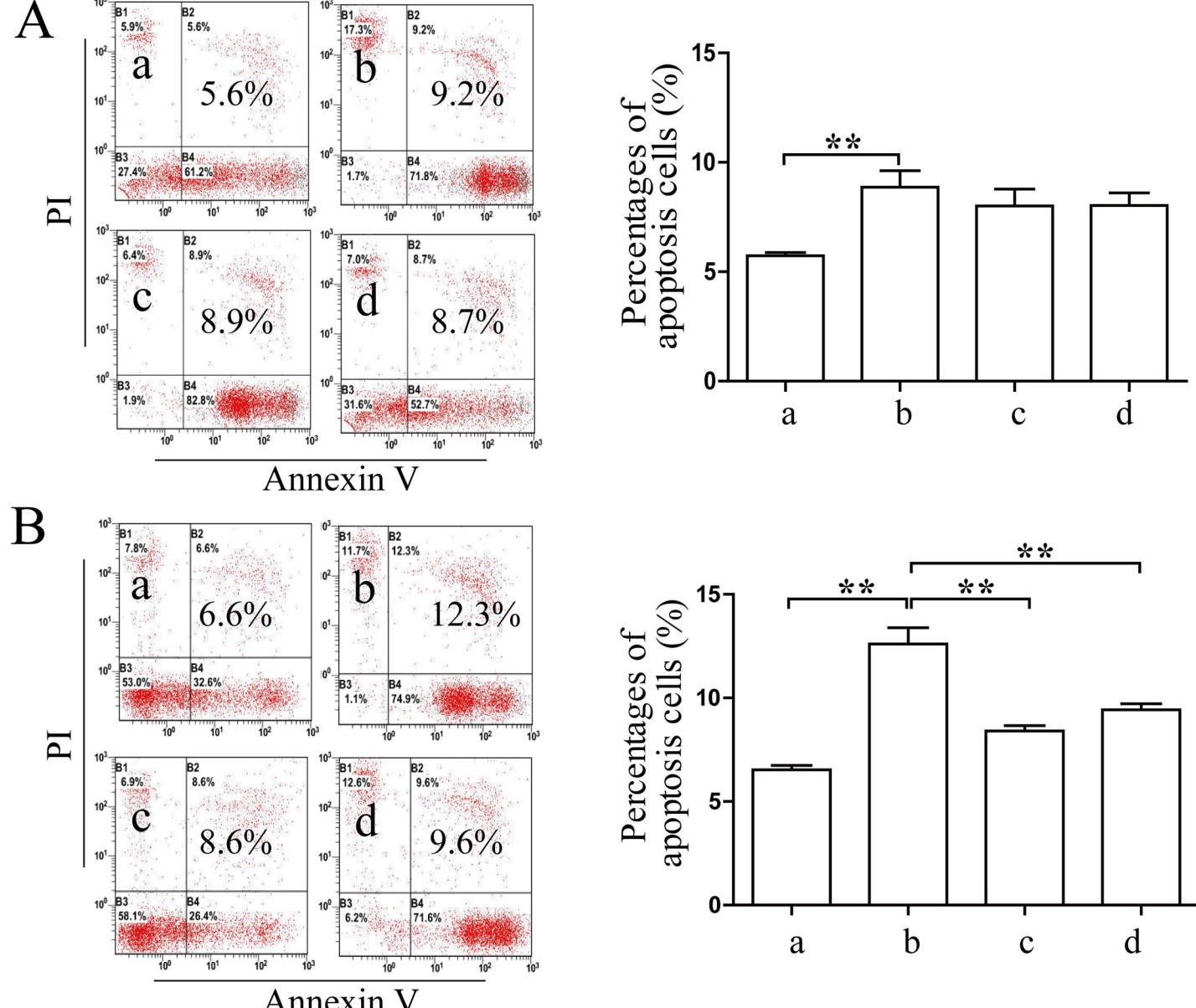

**Fig 2. Coptidis alkaloids attenuated the cell apoptosis of the IFN-γ-induced BMCs.** Coptidis alkaloids attenuated the cell apoptosis of BMCs induced by IFN-γ after 12 h (A) and 24 h (B) of treatment. The results are presented in a bar chart. a, Normal group; b, Model group; c, Coptidis alkaloid (100 μg/mL) group; d, Coptidis alkaloid (250 μg/mL) group. Data are presented as the mean ± SD, n = 3. $^{*}P<0.05$ and$^{**}P<0.01$.

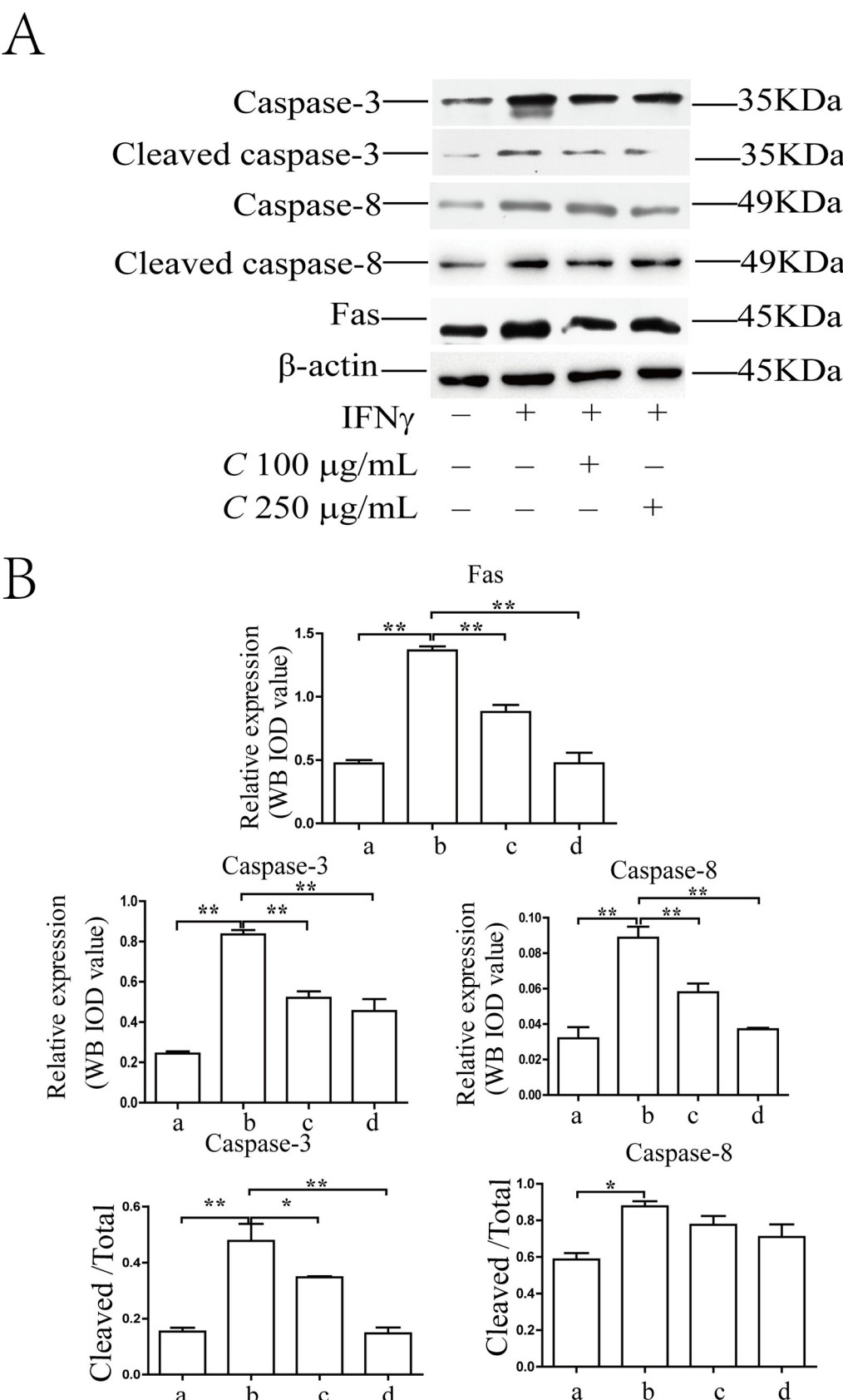

**Fig 3. Interference in the protein expression of key molecules in the Fas apoptotic signaling pathway in BMCs induced by IFN-γ after 12 h of Coptidis alkaloid treatment.** The results are presented in a bar chart. a, Normal group; b, Model group; c, Coptidis alkaloid (100 μg/mL) group; and d, Coptidis alkaloid (250 μg/mL) group. Data are presented as the mean ± SD, n = 3. *$P<0.05$, **$P<0.01$.

in the treatment group at 250 μg/mL was decreased more significantly than the other groups were after 24 h of treatment, showing an obvious dose-time effect relationship.

## The protein levels of key molecules in the Jak/Stats signaling pathway after Coptidis alkaloid treatment

As shown in Fig 11, the protein levels of Stat-1, Stat-3, phospho-Stat-1 and phospho-Stat-3 in the model groups were significantly higher than those in the normal groups ($P<0.01$ or $P<0.05$). *Coptidis alkaloid* treatment downregulated the abnormally high expression of Stat-1 and Stat-3 ($P<0.05$ or $P<0.01$, compared with the model groups). The aberrant phosphorylation activations of Stat1, and Stat3 were decreased significantly in the treatment groups. There are no differences in dose and time dependence.

## Discussion

*Coptis chinensis* is an herb of TCM known for its antibacterial, antioxidative and anti-inflammatory functions. Pharmacological research has confirmed that berberine and Coptisine have regulatory functions in the activation of the NF-κB, MAPK or PI3K/Akt signaling pathways, contributing to the inhibition of the inflammatory response induced by pro-inflammatory cytokines [15, 16]. Magnoflorine could regulate the differentiation of neutrophils and T cells resulting in an anti-inflammatory effect [17]. Jatrorrhizine, epiberberine and berberine had inhibitory functions on cell apoptosis through antioxidative activation or by intervening in the inducible nitric monoxide synthase system [10]. Analysis of the aqueous extract from *Coptis chinensis* showed that the dominant ingredient was berberine. There were also small amounts of other alkaloids in the aqueous extract, including magnoflorine, jatrorrhizine, epiberberine and coptisineis. These findings suggested that the Coptidis alkaloids extracted from *Coptis chinensis* by water decoction had regulatory effects on the inflammatory response and cell apoptosis of BMCs induced by pro-inflammatory cytokines.

Abnormal IFN-γ production plays an important role in the pathogenesis of the immune-mediated destruction of hematopoiesis. IFN-γ can activate T cell cellular receptors and the Fas receptor [18]. Trimerization of the Fas receptor activates caspases, and ultimately promotes cell apoptosis [19]. The Fas receptor trimerizes and induces apoptosis through a cytoplasmic domain called the Fas-associated death domain (FADD) [20]. FADD maintains a DED containing procaspase-8 protein in an inactive state. Procaspase-8 is proteolytically activated to produce, which activates the downstream effector Caspase-3. In the nucleus, activated Caspase-3 cleaves the catalytic subunit of the DNA-dependent protein kinase and releases the cleaved fragments to the cytosolic compartment, inducing cell apoptosis [21]. Our results indicated that Coptidis alkaloids in the aqueous extract of *Coptis chinensis* could decrease the expression of Fas in BMCs induced by IFN-γ, and it could inhibit the activation of Caspase-3, and Caspase-8 through cleavage. For these reasons, we considered that the aqueous extract of *Coptis chinensis* could attenuate the apoptosis of BMCs induced by IFN-γ through inhibiting the activation of the Fas apoptotic signal pathway.

Antigen-presenting cells present antigens to T lymphocytes, triggering the activation and proliferation of T cells. Signaling lymphocytic activation molecule (SLAM)-associated protein (SAP) binds to Fyn and regulates the activity of SLAM. When SAP stops binding to Fyn, the

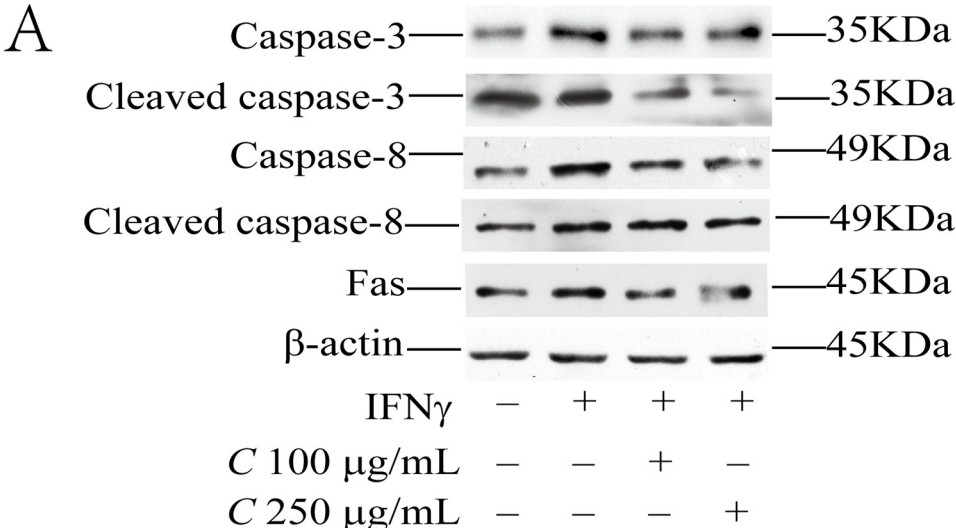

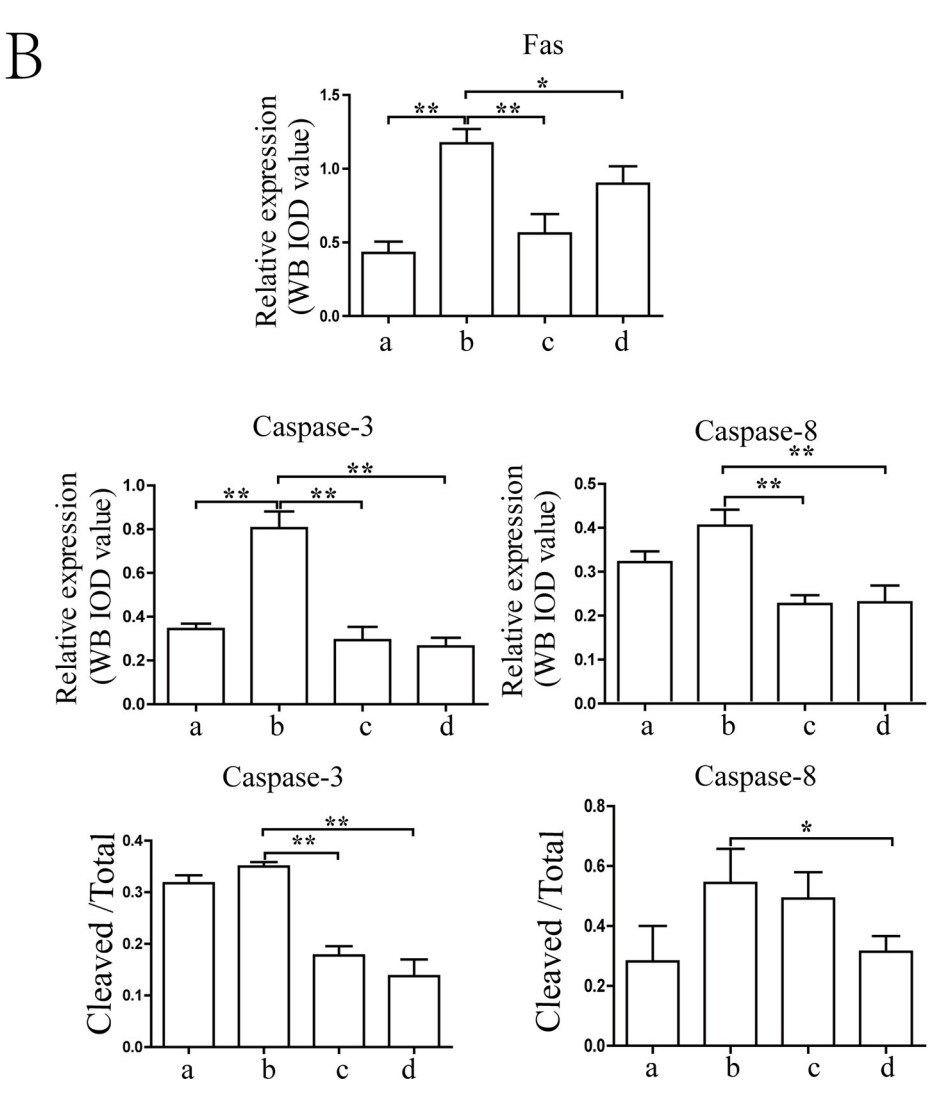

**Fig 4. Intervention in the protein expression of key molecules in the Fas apoptotic signaling pathway in BMCs induced by IFN-γ after 24 h of Coptidis alkaloid treatment.** The results are presented in a bar chart. a, Normal group; b, Model group; c, Coptidis alkaloid (100 μg/mL) group; and d, Coptidis alkaloid (250 μg/mL) group. Data are presented as the mean ± SD, n = 3. *P<0.05, and **P<0.01.

SLAM cascade is activated [22]. T-bet binds to the promoter region of IFN-γ and induces its expression. Constitutive T-bet expression and low SAP levels are present in the pathophysiology of immune-mediated bone marrow failure. IFN-γ promotes the secretion of interleukin-2 and contributes to the polyclonal expansion of T cells [23]. In the present study, we found that Coptidis alkaloids in the aqueous extract of *Coptis chinensis* could decrease the expression and distribution of T-bet in BMCs induced by IFN-γ, reduce the production of IFN-γ, and inhibit the activation of T cells.

CD4[+] T cells (including Th1, Th2, Th17 and Treg cells) have potential roles in the pathogenesis of immune-mediated BM failure [24]. Compared to other CD4+ T cells, a higher proportion of Th1 cells produce IFN-γ and IL-2, while Th2 cells do not pursue a similar regulatory function; thus, there is a shift of the IFN-γ/ IL-4 ratio, resulting in the immune-mediated destruction of hematopoietic stem/progenitor cells [25]. Activated Th1 cells secrete IFN-γ and TNFα and destroy hematopoietic colony formation [26]. Tregs play key roles in autoimmunity. The proportion of Tregs correlates with the severity of immune-mediated BM failure. Tregs have a suppressive function on the activation of effector T cells, including IFN-γ

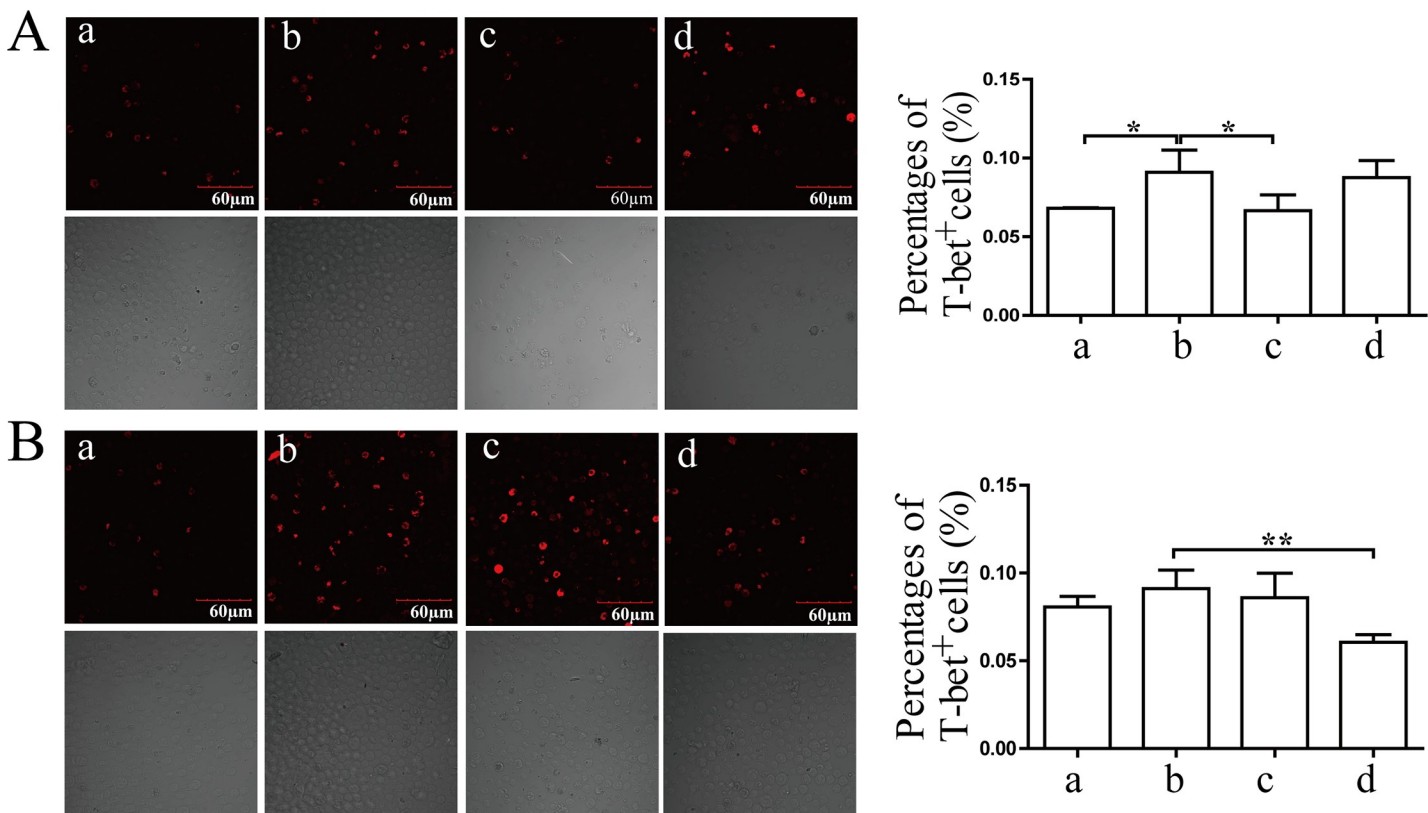

**Fig 5.** The expression and distribution of T-bet in BMCs induced by IFN-γ after 12 h (A) or 24 h (B) of treatment. The scale bar corresponds to 60 μm throughout. The quantified results are presented in a bar chart. a, Normal group; b, Model group; c, Coptidis alkaloid (100 μg/mL) group; and d, Coptidis alkaloid (250 μg/mL) group. Data are presented as the mean ± SD, n = 3. *P<0.05 and **P<0.01.

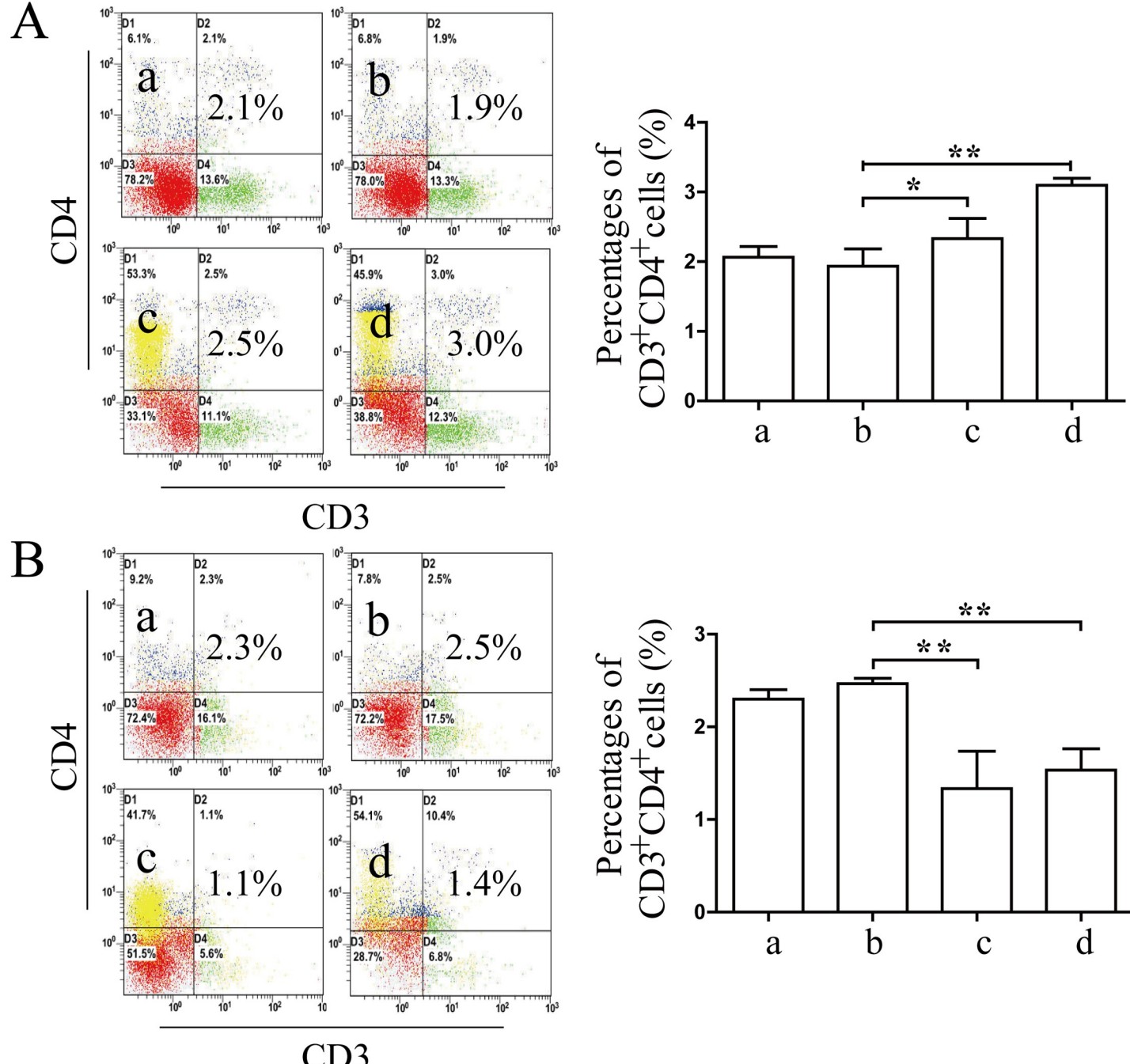

**Fig 6.** Coptidis alkaloids regulated the proliferation of CD3+CD4+ T cells in BMCs after 12 h (A) or 24 h (B) of treatment. The results are presented in a bar chart. a, Normal group; b, Model group; c, Coptidis alkaloid (100 μg/mL) group; and d, Coptidis alkaloid (250 μg/mL) group. Data are presented as the mean ± SD, n = 3. *P<0.05 and**P<0.01.

production [27]. Th17 cells produce proinflammatory cytokines, chemokines, growth factors and adhesion molecules that augment neutrophil accumulation. The differentiation and expansion of Th17 cells correlate with the depletion of Treg function in BM failure [28]. Suppression of the expansion of Th17 cells by treatment with an anti-IL-17 antibody could increase the proportion of Treg cells in a murine model of immune-mediated bone marrow

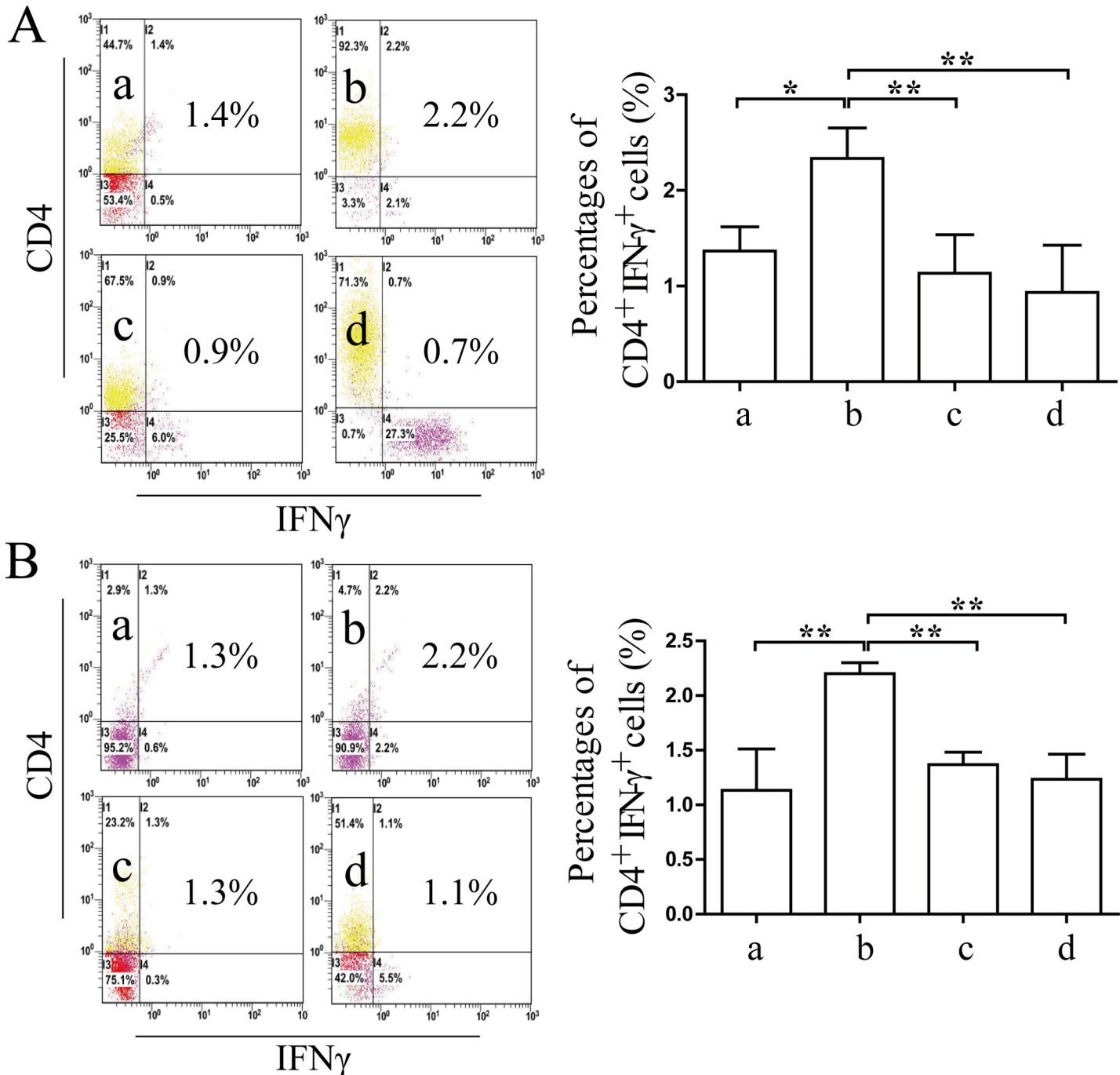

**Fig 7.** The percentages of CD4+IFN-γ+ T cells in BMCs after 12 h (A) or 24 h (B) of Coptidis alkaloid treatment. The results are presented in a bar chart. a, Normal group; b, Model group; c, Coptidis alkaloid (100 μg/mL) group; and d, Coptidis alkaloid (250 μg/mL) group. Data are presented as the mean ± SD, n = 3. $^*P<0.05$ and $^{**}P<0.01$.

failure, downregulate the production of IFN-γ levels, and reduce the severity of bone marrow failure [29]. Our results indicated that Coptidis alkaloid treatment could interfere with the differentiation of CD4+ T cells, deplete the expansion of Th1 and Th17 cells, augment the proportions of Th2 and Treg cells, and reduce the severity of bone marrow failure mediated by immune destruction.

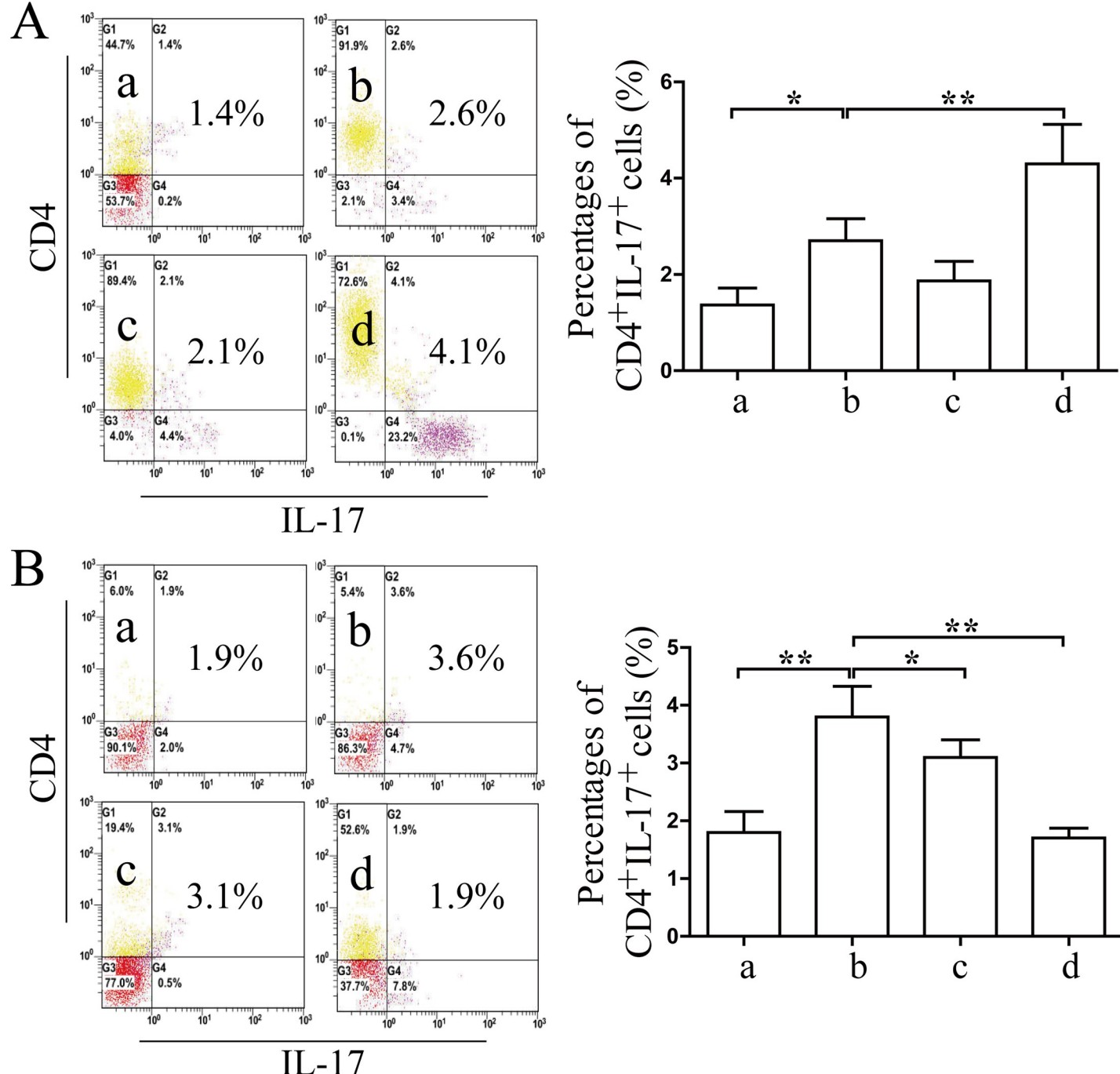

**Fig 8.** The percentages of CD4[+]IL-17[+] T cells in BMCs after 12 h (A) or 24 h (B) of Coptidis alkaloid treatment. The results are presented in a bar chart. a, Normal group; b, Model group; c, Coptidis alkaloid (100 μg/mL) group; and d, Coptidis alkaloid (250 μg/mL) group. Data are presented as the mean ± SD, n = 3. [*]$P<0.05$ and [**]$P<0.01$.

IFN-γ can also activate interferon regulatory factor 1, reduce the transcription of genes and inhibit the cell cycle of BMCs through the activation of the Jak/Stat signaling pathway. IFN-γ binds to its receptor, induces the activation of interferon response factors, and leads to alterations in gene transcription through activation of Jak/Stat signaling. The major transcription

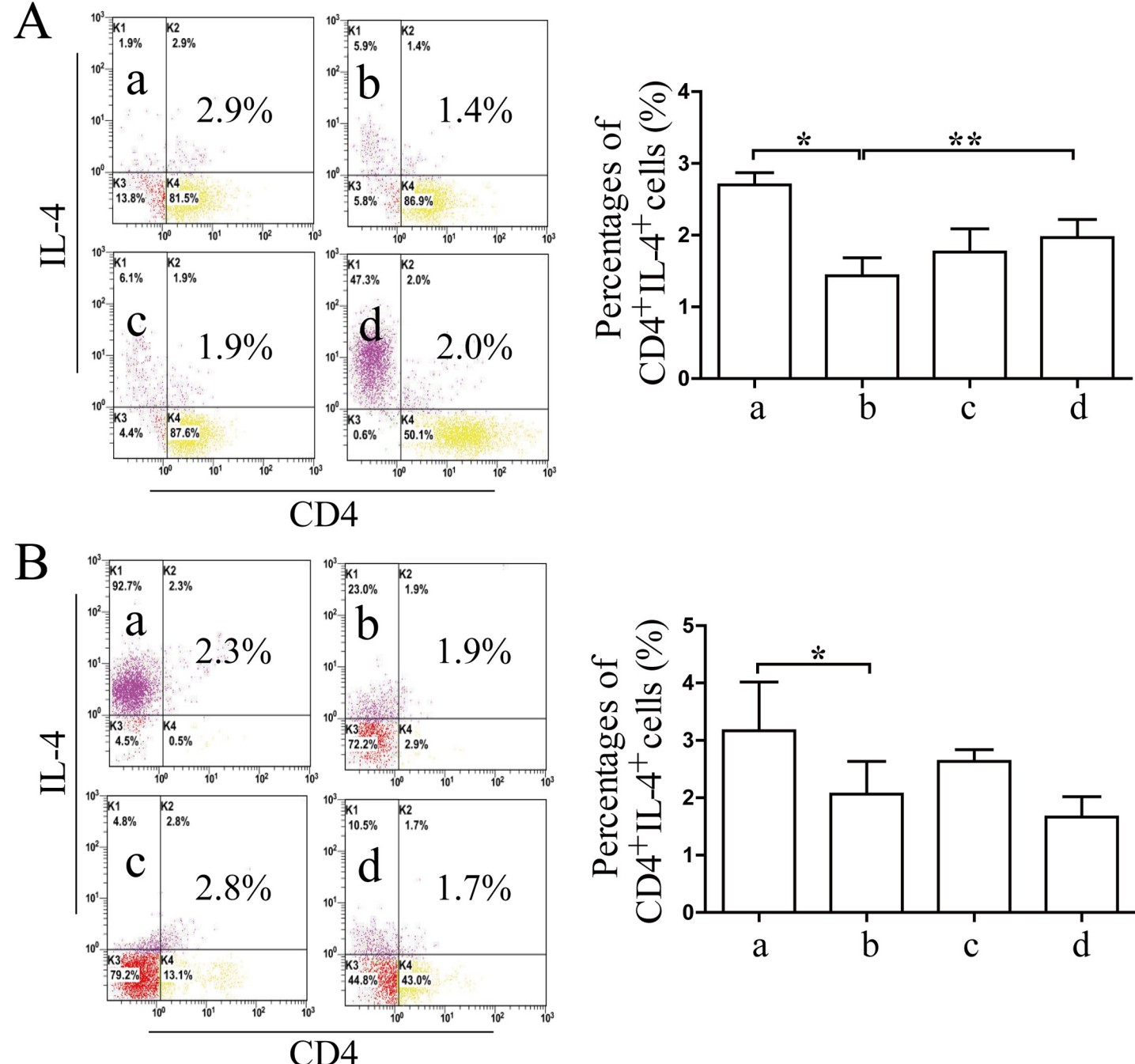

**Fig 9.** The percentages of CD4+IL-4+ T cells in BMCs after 12 h (A) or 24 h (B) of Coptidis alkaloid treatment. The results are presented in a bar chart. a, Normal group; b, Model group; c, Coptidis alkaloid (100 μg/mL) group; and d, Coptidis alkaloid (250 μg/mL) group. Data are presented as the mean ± SD, n = 3. *P<0.05 and **P<0.01.

factor connected with IFN-γ receptor signaling is Stat1 [30, 31]. IFN-γ signaling also occurs through Stat3 activation [32]. Phosphorylated and dimerized Stat3 enters the nucleus through as a result of binding or indirect regulation by Jak signaling, and it promotes the transcriptional expression of target genes to promote various cellular processes [33]. Our results also

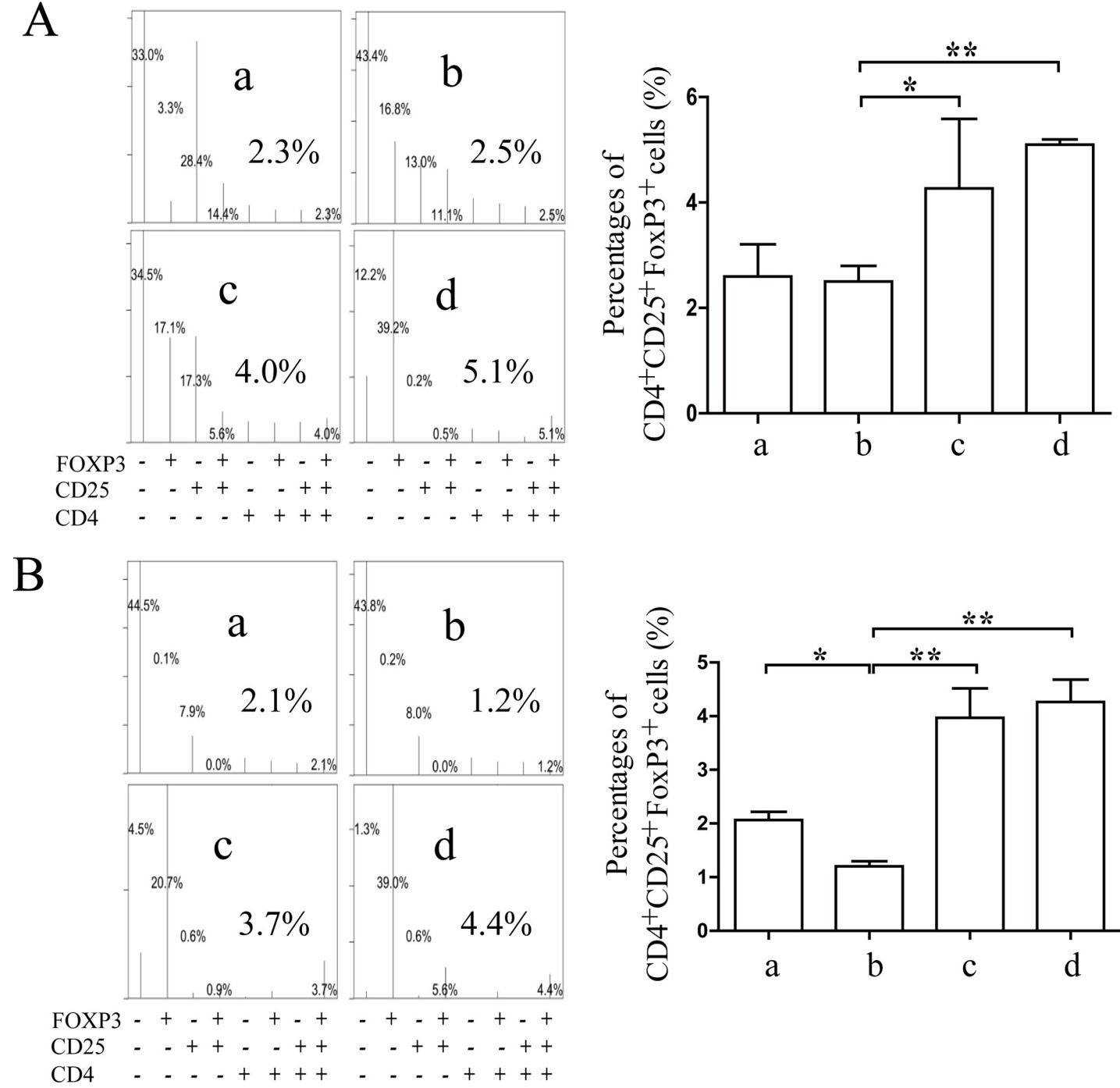

**Fig 10.** The percentages of CD4[+]CD25[+]FOXP3[+] T cells in BMCs after 12 h (A) or 24 h (B) of Coptidis alkaloid treatment. The results are presented in a bar chart. a, Normal group; b, Model group; c, Coptidis alkaloid (100 μg/mL) group; and d, Coptidis alkaloid (250 μg/mL) group. Data are presented as the mean ± SD, n = 3. *P<0.05 and **P<0.01.

showed that Coptidis alkaloids could inhibit the expression and activation of Stat1 and Stat3 signaling and restrain the reduction of cell cycling and cell death by apoptosis induced by IFN-γ-IRF-Jak/Stat signaling.

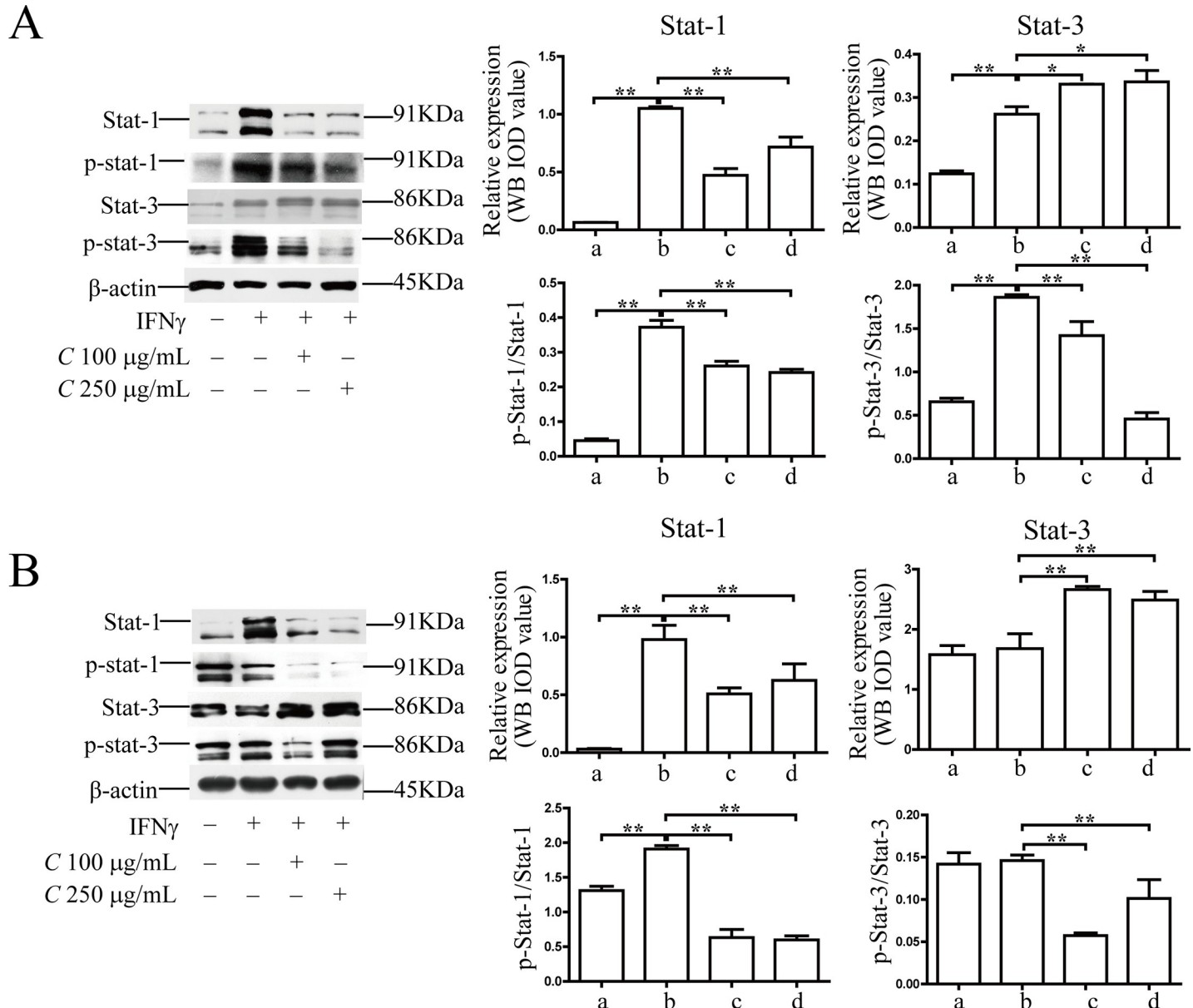

**Fig 11.** The protein levels of key molecules in the Jak/Stat signaling pathway in BMCs induced by IFN-γ after 12 h (A) or 24 h (B) of treatment. The results are presented in a bar chart. a, Normal group; b, Model group; c, Coptidis alkaloid (100 μg/mL) group; and d, Coptidis alkaloid (250 μg/mL) group. Data are presented as the mean ± SD, n = 3. *$P<0.05$ and **$P<0.01$.

## Conclusions

In summary, our study indicated that Coptidis alkaloids in the aqueous extract from *Coptis chinensis* could attenuate the immunosuppressive function of Treg cells by regulating the proliferation and differentiation of effector CD4[+] T cells in BMCs induced by IFN-γ. Coptidis alkaloids could also inhibit the activation of IFN-γ-IRF-Jak/Stat signaling, contribute to the resolution of aberrant immune responses, and attenuate immune-mediated destruction of BMCs in immune-mediated BM failure. This therapeutic efficacy appears to be time and dose-dependent. Coptidis alkaloid treatment at a considerably higher dose for a longer treatment

time could appear to have a better effect than what was tested here. Our results confirmed that Coptidis alkaloids (the dominant ingredient in aqueous extract from *Coptis chinensis*) were the effector substances of *Coptis chinensis* directing the immunosuppressive function on immune-mediated BM failure.

## Supporting information

**S1 Fig.**
(TIFF)

**S1 File.**
(PDF)

## Acknowledgments

The authors thank Dr Shuyan Ma for technical assistance.

## Author Contributions

**Conceptualization:** Changzhi Wang, Juan Liu.

**Data curation:** Jinyu Li, Changzhi Wang, Peiying Deng, Meiyier Huandike.

**Formal analysis:** Peiying Deng, Meiyier Huandike.

**Funding acquisition:** Limin Chai.

**Investigation:** Huijie Zhang.

**Methodology:** Huijie Zhang.

**Project administration:** Limin Chai.

**Software:** Xiaoying Meng, Juan Liu.

**Supervision:** Hening Chen.

**Validation:** Hening Chen, Jie Wei, Limin Chai.

**Writing – review & editing:** Jinyu Li, Xiaoying Meng, Jie Wei, Limin Chai.

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
