## [Decision Letter · Decision Letter 0]

1 May 2020

PONE-D-20-06039

The Coptidis alkaloids extracted from Coptis chinensis Franch attenuates the IFN-γ-induced immune destruction in bone marrow cells

PLOS ONE

Dear Dr. Chai,

Thank you for submitting your manuscript to PLOS ONE. After careful consideration, we feel that it has merit but does not fully meet PLOS ONE’s publication criteria as it currently stands. Therefore, we invite you to submit a revised version of the manuscript that addresses the points raised during the review process.

In addition to address the concerns raised by the reviewers, please have the English polished so that readers can better understand the study.

We would appreciate receiving your revised manuscript by Jun 15 2020 11:59PM. To enhance the reproducibility of your results, we recommend that if applicable you deposit your laboratory protocols in protocols.io, where a protocol can be assigned its own identifier (DOI) such that it can be cited independently in the future. For instructions see: http://journals.plos.org/plosone/s/submission-guidelines#loc-laboratory-protocols

We look forward to receiving your revised manuscript.

Kind regards,

Xing-Ming Shi, Ph.D

Academic Editor

PLOS ONE

Journal Requirements:

1. PLOS ONE now requires that authors provide the original uncropped and unadjusted images underlying all blot or gel results reported in a submission’s figures or Supporting Information files. This policy and the journal’s other requirements for blot/gel reporting and figure preparation are described in detail at https://journals.plos.org/plosone/s/figures#loc-blot-and-gel-reporting-requirements and https://journals.plos.org/plosone/s/figures#loc-preparing-figures-from-image-files. When you submit your revised manuscript, please ensure that your figures adhere fully to these guidelines and provide the original underlying images for all blot or gel data reported in your submission. See the following link for instructions on providing the original image data: https://journals.plos.org/plosone/s/figures#loc-original-images-for-blots-and-gels.

2, In the Methods, please provide the precise part of Coptis chinensis Franch used in this work (roots, stems, leaves, etc).

- We note that  Coptis chinensis Franch was obtained from Beijing Xidan Pharmaceutical Co, Ltd, China. As such, in the Methods, please provide the product number and lot number.

- We noticed minor instances of text overlap with the following previous publications from your group, which need to be addressed:

https://bmccomplementmedtherapies.biomedcentral.com/articles/10.1186/s12906-019-2781-4

https://journals.plos.org/plosone/article?id=10.1371%2Fjournal.pone.0180417

In your revision please ensure you cite all your sources (including your own works), and quote or rephrase any duplicated text outside the methods section. Further consideration is dependent on these concerns being addressed.

3. Thank you for including the following funding information within your acknowledgements; "This work was supported by the State Administration of Traditional Chinese Medicine, National Chinese Medicine Clinical Research Base, Research special [grant number: JDZX2015191]."

Reviewers' comments:

Reviewer's Responses to Questions

**Comments to the Author**

1. Is the manuscript technically sound, and do the data support the conclusions?

Reviewer #1: Yes

Reviewer #2: Yes

2. Has the statistical analysis been performed appropriately and rigorously? 

Reviewer #1: Yes

Reviewer #2: Yes

3. Have the authors made all data underlying the findings in their manuscript fully available?

Reviewer #1: Yes

Reviewer #2: Yes

4. Is the manuscript presented in an intelligible fashion and written in standard English?

Reviewer #1: Yes

Reviewer #2: Yes

5. Review Comments to the Author

Reviewer #1: This is a well-written study in which the effect of coptidis alkaloids on IFN-gamma induced destruction in mice bone marrow cells. All experiments and analyses are performed on high technical standards; data are adequately presented and discussed.

I have only two minor points:

1. What are bone marrow cells? Which type of cells are present there? This should be shown by Authors, because several cell types could mediate the effect of alkaloids and IFN-gamma.

Authors used ANOVA statistics, but how they proved the normal distribution of data, which is a pre-requisite for this statistical method?

Reviewer #2: Dear Limin Chai, M.D.

Thank you for your manuscript submission.

Your manuscript titled “The Coptidis alkaloids extracted from Coptis chinensis Franch attenuates the IFN-γinduced immune destruction in bone marrow cells ” has been reviewed and here we put forward some suggestions so that you can make essential modifications before you submit it next time.

1)Some typos or grammatical errors should be corrected. (e.g. The fifth line of the Cell treatment part, ‘...... cells were harvest’. The first line of Discussion part, ‘Coptis chinensis are known for its antibacterial ......’.) We would like you to read your whole manuscript again to make sure there are no mistakes like these.

2)We would like you to provide more reliable information in figure 5 about your immunofluorescence test. Pictures of DAPI staining should be provided to confirm the total number and the location of cells in each visual field.

3)We would like you to provide the all the pictures of the whole membranes of your WB tests (including protein markers) in the supplementary figure to ensure the results are reliable.

4)We suggest that animal experiments in vivo should be added to further demonstrate your point of view. For example, IHC or HE staining might be applied to analyze changes of the IFN-γ-induced bone marrow tissues after being treated with coptis or not in different time.

These issues have to be dealt with before this manuscript get published. We hope our suggestions help you to modify your article to address these issues.

Yours sincerely

Dr Huang

6. PLOS authors have the option to publish the peer review history of their article (what does this mean?). If published, this will include your full peer review and any attached files.

Reviewer #1: No

Reviewer #2: No

---

## [Author Response · Author response to Decision Letter 0]

23 Jun 2020

Journal Requirements:

1. PLOS ONE now requires that authors provide the original uncropped and unadjusted images underlying all blot or gel results reported in a submission’s figures or Supporting Information files. This policy and the journal’s other requirements for blot/gel reporting and figure preparation are described in detail at https://journals.plos.org/plosone/s/figures#loc-blot-and-gel-reporting-requirements and https://journals.plos.org/plosone/s/figures#loc-preparing-figures-from-image-files. When you submit your revised manuscript, please ensure that your figures adhere fully to these guidelines and provide the original underlying images for all blot or gel data reported in your submission. See the following link for instructions on providing the original image data: https://journals.plos.org/plosone/s/figures#loc-original-images-for-blots-and-gels.

Response: We have provided the original uncropped gel image in the uploaded files.

2, In the Methods, please provide the precise part of Coptis chinensis Franch used in this work (roots, stems, leaves, etc).

- We note that Coptis chinensis Franch was obtained from Beijing Xidan Pharmaceutical Co, Ltd, China. As such, in the Methods, please provide the product number and lot number.

- We noticed minor instances of text overlap with the following previous publications from your group, which need to be addressed:

https://bmccomplementmedtherapies.biomedcentral.com/articles/10.1186/s12906-019-2781-4

https://journals.plos.org/plosone/article?id=10.1371%2Fjournal.pone.0180417

In your revision please ensure you cite all your sources (including your own works), and quote or rephrase any duplicated text outside the methods section. Further consideration is dependent on these concerns being addressed.

Response: Thanks for this comment. We have provided the precise part lot number of Coptis chinensis Franch in the Methods of revised manuscript. We also have removed and re-described the relevant contents which overlap with our previously published works in the revised manuscript.

3. Thank you for including the following funding information within your acknowledgements; "This work was supported by the State Administration of Traditional Chinese Medicine, National Chinese Medicine Clinical Research Base, Research special [grant number: JDZX2015191]."

Response: We have removed the funding-related text and Funding Statement in the revised manuscript. This work was supported by Research special for National Chinese Medicine Clinical Research Base of State Administration of Traditional Chinese Medicine of the People’s Republic of China (Grant number: JDZX2015191, http://www.satcm.gov.cn/). The funder had no role in study design, data collection and analysis, decision to publish, or preparation of the manuscript. We have uploaded the information of funding on the revised files.

Reviewers' comments:

Reviewer #1: This is a well-written study in which the effect of coptidis alkaloids on IFN-gamma induced destruction in mice bone marrow cells. All experiments and analyses are performed on high technical standards; data are adequately presented and discussed.

I have only two minor points:

1. What are bone marrow cells? Which type of cells are present there? This should be shown by Authors, because several cell types could mediate the effect of alkaloids and IFN-gamma.

Response: We obtained bone marrow cells from the tibias, femurs and humeris of BALB/c mice. We focused on the apoptotic BMCs and the proliferation and differentiation of helper T cells and regulatory T cells induced by IFN-γ in the bone marrow microenvironment treated by alkaloids extracted from Coptis chinensis Franch. From the above consideration, we did not separate and identify the cellular constituent of bone marrow cells in this study.

2. Authors used ANOVA statistics, but how they proved the normal distribution of data, which is a pre-requisite for this statistical method?

Response: Thank you for this comment. We used the Shapiro-Wilk test method for data normality testing. We have added instructions about data normality testing in the statistical method section of the revised manuscript.

Reviewer #2: 

Your manuscript titled “The Coptidis alkaloids extracted from Coptis chinensis Franch attenuates the IFN-γinduced immune destruction in bone marrow cells ” has been reviewed and here we put forward some suggestions so that you can make essential modifications before you submit it next time.

1. Some typos or grammatical errors should be corrected. (e.g. The fifth line of the Cell treatment part, ‘...... cells were harvest’. The first line of Discussion part, ‘Coptis chinensis are known for its antibacterial ......’.) We would like you to read your whole manuscript again to make sure there are no mistakes like these.

Response: According to the reviewer’s comment, we have corrected the grammatical errors and the unclear or awkwardly worded sentences. And then, the manuscript has been edited by American Journal experts.

2. We would like you to provide more reliable information in figure 5 about your immunofluorescence test. Pictures of DAPI staining should be provided to confirm the total number and the location of cells in each visual field.

Response: Thank you for comment on this important concern. According to your comment, we added the pictures obtained in bright field corresponding to the immunofluorescence pictures in Figure 5. The pictures in bright field also could confirm the total number and the location of cells in the visual field. Because of the coVID-19 outbreak, we cannot supplement the relevant experiments of DAPI staining. We hope that this supplement will meet with approval.

3. We would like you to provide the all the pictures of the whole membranes of your WB tests (including protein markers) in the supplementary figure to ensure the results are reliable.

Response: We have provided the original uncropped gel image in the uploaded files. We used a pre-dyed marker in the WB tests. The marker could not be displayed on the films. For this reason, we provided the original uncropped gel image without protein markers.

4. We suggest that animal experiments in vivo should be added to further demonstrate your point of view. For example, IHC or HE staining might be applied to analyze changes of the IFN-γ-induced bone marrow tissues after being treated with coptis or not in different time.

Response: Thank you for this comment. The animal experiments for modified Danggui Buxue Tang (Radix astragali, Radix Angelicae sinensis, and Coptis chinensis Franch) have been done in our previous work. The experimental results have been published in our papers (1 Jingwei Zhou, Xue Li, Peiying Deng, et al. Chinese Herbal Formula, modified Danggui Buxue Tang, attenuates apoptosis of hematopoietic stem cells in immune mediate aplastic anemia mouse model, Journal of Immunology Research, 2017, 2017(2017): 9786972; 2 Peiying Deng, Xue Li,Yi Wei, et al. The herbal decoction modified Danggui Buxue Tang attenuates immune-mediated bone marrow failure by regulating the differentiation of T lymphocytes in an immune-induced aplastic anemia mouse model, PLOS ONE, 2017, 12(7): e0180417). In this study, we focus on the exact mechanism of Coptis chinensis Franch for regulating the apoptosis of BMCs and the proliferation and differentiation of helper T cells and regulatory T cells induced by IFN-γ in the bone marrow microenvironment in vitro.

---

## [Decision Letter · Decision Letter 1]

8 Jul 2020

Coptidis alkaloids extracted from Coptis chinensis Franch attenuate IFN-γ-induced destruction of bone marrow cells

PONE-D-20-06039R1

Dear Dr. Chai,

We’re pleased to inform you that your manuscript has been judged scientifically suitable for publication and will be formally accepted for publication once it meets all outstanding technical requirements.

Kind regards,

Xing-Ming Shi, Ph.D

Academic Editor

PLOS ONE

Additional Editor Comments (optional):

Reviewers' comments:

Reviewer's Responses to Questions

**Comments to the Author**

1. If the authors have adequately addressed your comments raised in a previous round of review and you feel that this manuscript is now acceptable for publication, you may indicate that here to bypass the “Comments to the Author” section, enter your conflict of interest statement in the “Confidential to Editor” section, and submit your "Accept" recommendation.

Reviewer #1: All comments have been addressed

Reviewer #2: (No Response)

2. Is the manuscript technically sound, and do the data support the conclusions?

Reviewer #1: Yes

Reviewer #2: Yes

3. Has the statistical analysis been performed appropriately and rigorously? 

Reviewer #1: Yes

Reviewer #2: Yes

4. Have the authors made all data underlying the findings in their manuscript fully available?

Reviewer #1: Yes

Reviewer #2: Yes

5. Is the manuscript presented in an intelligible fashion and written in standard English?

Reviewer #1: Yes

Reviewer #2: Yes

6. Review Comments to the Author

Reviewer #1: (No Response)

Reviewer #2: We are so glad that you and your team made some modification to your manuscript according to the comments. We believe that your manuscript can now be accepted and published in the journal of PLOS ONE

7. PLOS authors have the option to publish the peer review history of their article (what does this mean?). If published, this will include your full peer review and any attached files.

Reviewer #1: No

Reviewer #2: No

---

## [Editor Report · Acceptance letter]

13 Jul 2020

PONE-D-20-06039R1 

 Coptidis alkaloids extracted from Coptis chinensis Franch attenuate IFN-γ-induced destruction of bone marrow cells 

Dear Dr. Chai:

I'm pleased to inform you that your manuscript has been deemed suitable for publication in PLOS ONE. Congratulations! Your manuscript is now with our production department. 

Kind regards, 

on behalf of

Dr Xing-Ming Shi 

Academic Editor

PLOS ONE